# Results of Utilizing Cerclage Wires in the Management of Intraoperative Vancouver B1 Fractures in Primary Total Hip Arthroplasties: A Retrospective Cohort Investigation into Clinical and Radiographic Outcomes

**DOI:** 10.3390/jcm13030819

**Published:** 2024-01-31

**Authors:** Caterina Rocchi, Vincenzo Di Matteo, Katia Chiappetta, Guido Grappiolo, Mattia Loppini

**Affiliations:** 1Department of Biomedical Sciences, Humanitas University, Pieve Emanuele, 20090 Milan, Italy; caterina.rocchi@st.hunimed.eu (C.R.); drvincenzodimatteo@gmail.com (V.D.M.); 2Adult Reconstruction and Joint Replacement Service, Division of Orthopaedics and Traumatology, Fondazione Policlinico Universitario Agostino Gemelli IRCCS, Largo Agostino Gemelli 8, 00168 Roma, Italy; 3Fondazione Livio Sciutto Onlus, Campus Savona, Università degli Studi di Genova, 17100 Savona, Italy; guido.grappiolo@mac.com; 4IRCCS Humanitas Research Hospital, Rozzano, 20089 Milan, Italy; katia.chiappetta@humanitas.it

**Keywords:** hip, knee, periprosthetic femur fractures, total hip arthroplasty

## Abstract

Background: Due to an increase in total hip arthroplasties (THAs), the incidence of periprosthetic hip fractures (PPHFs) is forecast to rise considerably in the next decades, with Vancouver B1 fractures (VB1) accounting for one third of total cases. Femur fixation with cerclages (with or without screws) is considered the current treatment option for intraoperative VB1. Methods: The study retrospectively includes data from patients who developed VB1 PPHFs during THAs from 3 December 2020 to 30 November 2022. The primary outcome of this study was to identify the reintervention-free survival rate. The secondary aim was to determine clinical and radiographic assessment at follow-up, based on Harris hip score (HHS) and limb length discrepancy (LLD). Results: Thirty-seven patients with a mean age of 60.03 ± 15.49 (22 to 77) years old were included. Overall, the Kaplan–Meier analysis estimated a reoperation-free survival rate of 99% (CI 95%) at 6 months. The mean limb length discrepancy (LLD) improved from −3.69 ± 6.07 (range −27.9 to 2.08) mm to 0.10 ± 0.67 (range −1.07 to 1.20) mm. The mean HHS improved from 42.72 ± 14.37 (range 21.00–96.00) to 94.40 ± 10.32 (range 56.00–100.00). Conclusions: The employment of cerclage wires represents an effective strategy for handling intraoperative VB1 fractures. Level III retrospective cohort study.

## 1. Introduction

Within a demographic characterized by an increasing prevalence of older individuals, the demand for total hip arthroplasties (THAs) is experiencing a progressive escalation on a global scale, and it is expected to augment from 1.8 million per year in 2015 to 2.8 (2.6–2.9) per year in 2050 [1]. The most important factors accounting for this tendency are the willingness of individuals to maintain a high-performance physical status and an active conduct throughout the whole lifespan and indications to perform THAs also in younger patients [2]. THAs are primarily effectuated on the elderly, who often experience the challenges of osteoporosis and diminished bone quality, in order to reduce pain and restore functionality in patients affected by osteoarthritis or other morbidities of the hip. A significant complication of THAs is represented by the development of periprosthetic hip fractures (PPHFs), that can originate both intraoperatively and postoperatively. They can be located both at level of the femur and/or on the acetabulum, with the latter being less common and more difficult to identify intraoperatively and on postoperative X-rays [3].

PPHFs place a significant burden on national healthcare systems worldwide due to the expenses correlated with increased lengths of hospital stays, along with costs related to surgery and implants (particularly in the case of revision arthroplasties) [4,5].

The elevation in PPHF incidence is due to the rising life expectancy among the population bearing prosthetic devices, which makes them more susceptible to develop bone fragility, undergo revisions or endure traumatic accidents [2]. It is extremely important to individuate and promptly address intraoperative PPHFs in order to avoid their propagation in postoperative settings. As a matter of fact, overlooking intraoperative periprosthetic femoral fractures may lead to early implant loosening and failure, causing damage to the surrounding soft tissues and increasing the risk of requiring revision surgery. This would expose patients to further intraoperative risks and place additional burden on healthcare systems.

PPHFs are classified based on the Vancouver classification system, developed by Duncan and Masri, which considers implant stability, fracture location and quality of the bone. Lesions are divided into type A (occurring at level of the femoral trochanters), type B (further classified into B1, B2 and B3) and type C [6]. VB1 fractures, which occur around a stable stem, account for almost one third of periprosthetic femoral fractures [7].

Multiple studies have been published in the past investigating the risk factors, postoperative morbidity and management of postoperative VB1 fractures; the latter can be treated surgically either with open reduction internal fixation (ORIF) for comminuted/long oblique fractures or by employing revision arthroplasty (RA) in cases of simple fractures/oblique fractures located at the tip of the stem [2]. Conversely, fractures occurring intraoperatively around a stable stem are typically entrusted to the surgeon’s expertise, unlike other types of fractures.

Intraoperative periprosthetic femoral fractures (IOPFFs), whose prevalence varies in the literature from 1.7% to 5.4% [8,9], are more often observed in female individuals and in patients over 65 years, and they occur 14 times more often around uncemented stems (0.3% for cemented vs. 5.4% for uncemented implants). [8,10] This could be linked to the fact that most IOPFFs occur during broaching of the femoral canal and press-fit implantation of uncemented stems [7], when the circumferential tension at the level of the proximal femur reaches the most elevated values [11]. Most IOPFFs occur during placement of the femoral component (60%) and involve the calcar (69%). In revisions, the risk of fracture is even higher, reaching 3.6% in cemented and 21% in uncemented cases.

Postoperative fracture risk, on the contrary, is independent of age or gender, but still seems to rise with uncemented stems [9,10]. However, findings from a more recent study suggests comparable occurrence of postoperative PPFs at 5 years for both cemented and uncemented implants, implying an ongoing controversy on this matter [12].

The most common sites of occurrence of IOPFFs are the femoral diaphysis, calcar and trochanteric regions.

The most important risk factors correlated with the development of IOPFFs, other than female gender, elderhood and cementless fixation, are the following: a diagnosis different to primary osteoarthritis, previous surgery on the same hip and employment of a direct lateral rather than a posterior approach [13,14]. When fractures around a stable stem (B1) occur intraoperatively, the femoral component is removed and cerclage wires are usually applied prior to stem reimplantation.

Previous investigations have delved into factors correlated to an increased risk of non-union of Vancouver B fractures treated with cerclage wiring, identifying as the most important risk factors associated with the occurrence of this complication patients’ age and female gender [15]. Conversely, the type of cerclage employed (metallic vs. iso-elastic cabling) does not seem to entail significant changes in non-union rates at one year; even though metallic wiring might to be associated with a higher risk of failure [16].

Effectively handling IOPFFs that occur during total hip replacement entails significant advantages for patients, enhancing both their quality of life and reoperation-free survival. However, in the current literature there are no indications of a gold standard for the specific treatment of VB1 fractures occurring during primary THAs.

The primary outcome of this study is determining the reintervention-free survival of patients affected by IOPFFs treated with cerclage wires (with or without screws) during primary THAs. The secondary aim, instead, is to analyze clinical and radiographic assessment at the time of follow-up, based on the Harris hip score (HHS), limb length discrepancy (LLD) and radiographic indicators of implant failure.

A prompt lesion identification and the employment of cerclage wires with or without screws represents an effective strategy for handling intraoperative VB1 fractures, correlating with excellent postoperative recovery and minimal occurrence of complications.

## 2. Patients and Methods

### 2.1. Protocol

The present retrospective observational cohort study employed medical file records of patients included in a registry of orthopedic surgical procedures. It includes data from patients who developed Vancouver type B1 PPHFs during THAs from 3 December 2020 to 30 November 2022 at the IRCCS Humanitas Research Hospital, Italy. All individual participants signed a written informed consent to perform the surgery and a written informed consent to be included in the registry of orthopedic surgical procedures within the scope of research and improvement of clinical practice. All the surgical procedures were executed by senior specialists with wide experience in joint replacement surgery. Patients were identified from hospital clinical records using International Classification of Diseases, Ninth Revision, Clinical Modification (ICD9-CM) procedure codes 81.51 (total hip replacement) and diagnostic code 996.43 (periprosthetic femoral fracture). Subsequently, a second analysis was conducted researching the word “stable” within surgical records.

Eligibility criteria included all patients aged above 18 years old who sustained an intraoperative PPHFs Vancouver type B1 during primary THAs (both cemented and uncemented) and were treated surgically with the placement of cerclage wires with or without screws.

Exclusion criteria for participants included the presence of complications arising from infections and malignancies, as these conditions could potentially elevate bone fragility due to the presence of infection and osteolytic lesions. Vancouver fractures type A, B2, B3 and C were excluded from this study, along with fractures that occurred postoperatively.

All the intraoperative fractures were classified based on the Vancouver classification system, which considers the site of fracture, implant stability and surrounding bone stock (Table 1).

Total hip arthroplasties were performed with both cemented and short or conventional uncemented stems.

An analysis of medical records allowed to obtain demographic details of the patients (sex and age at surgery), affected side (left, right or bilateral), preoperative diagnosis (primary vs. secondary osteoarthritis), fracture etiology, Vancouver type, stem fixation (cemented vs. uncemented), surgical strategy, stem type (straight or anatomical variant), operating time and length of stay (LOS). The 30-day readmission rate, 12-month reoperation rate, and mortality rate at 30 days and 12 months were examined. Reintervention at 6 months and 1 year was recorded.

Furthermore, implant complications (non-union, loosening, dislocation, infection, heterotopic ossification) were recorded within 90 days of surgery along with adverse events experienced by the patients, including hospital acquired pneumonia, atrial fibrillation (AF), myocardial infarction, venous thromboembolism (VTE), urinary tract infections (UTIs), surgical site infection and sciatic nerve injury.

A thorough examination of medical records was conducted during the study, encompassing clinical and surgical details, follow-up visits and radiographic evaluations of all patients. Functional and radiographical assessments were collected both preoperatively at hospital admission and postoperatively during follow-up visits. During postoperative follow-up visits, the examiners followed a standardized protocol encompassing both clinical testing of patients’ performance status and evaluation of X-rays of the pelvis and the hips.

A quantitative assessment of functional hip recovery was made by comparing the preoperative and postoperative values of the Harris Hip Score, a disease-specific questionnaire employed to measure clinical outcomes. The score combines both a questionnaire directed to patients (pain, walking support, presence of limp, walking distance, ability to walk on stairs, put on shoes and sit) and a clinical evaluation of the range of movement (ROM) of the hip [17].

Conversely, radiographic evaluation relied on a comparison of limb length discrepancy estimates before and after the surgical procedure, measured from the anteroposterior view.

#### 2.1.1. Surgical Technique

All the procedures were carried out by senior surgeons experienced in joint replacement surgery. All THAs were performed using a standard posterolateral approach and femur first technique with the patient in lateral decubitus [18]. The operations were preceded by routine radiographic work-up and digital templating. Many different designs of monoblock stems were employed. In this analysis, the management of VB1 fractures occurring intraoperatively, and in most cases around uncemented stems, was investigated.

All the fractures were identified after the definitive femoral stem implantation, and their location and extent, along with stem stability, were assessed intraoperatively. Fractures were classified as VB1 if the femoral stem remained placed in the proper position; stem stability prevented both implant subsidence and rotational instability, ensuring the vertical and lateral femoral offsets planned (the distance between the lesser trochanter and the center of rotation, and the greater trochanter and the center of rotation, respectively). It was viable to opt for the same stem size for replacement due to the preservation of bone stock. Afterwards, the stem was removed and fractures were treated with femur fixation by metal or iso-elastic cerclages with or without screws prior to definitive reimplantation of the femoral component, after having removed eventual bone fragments from the medullary canal. It is important to identify intraoperative fractures to avoid their future propagation in postoperative settings.

#### 2.1.2. Postoperative Rehabilitation

The rehabilitation process after THAs complicated by intraoperative VB1 fractures remained consistent with standard postoperative management for uncomplicated hip replacement surgeries. The recovery regimen began in the immediate postoperative period and was tailored to the specific needs and physical status of each patient. Early mobilization exercises, including soft activities for range of motion (ROM) recovery and ambulation with assistive devices, were initiated to prevent joint stiffness and muscle atrophy development. As the healing progressed, the physiotherapeutic program evolved to include strengthening exercises, balance training and activities aimed at restoring normal gait. Partial weightbearing was suggested for the first 3 to 4 weeks postoperatively. The primary goal of postoperative rehabilitation was to improve patients’ functional abilities, reduce pain and promote a safe return to daily activities. Close collaboration between the surgical team, physical therapists and patients has been essential to ensure a successful rehabilitation process and achieve the best possible outcomes in terms of mobility and quality of life.

### 2.2. Statistical Analysis

Data analysis was performed with STATA 17, considering a *p*-value under 0.05 as significant.

Discrete variables were described as numbers and percentages, whereas for continuous variables the mean value and standard deviation were obtained, with range if necessary. Adherence of continuous variables to a Gaussian distribution was checked with the Shapiro–Wilk test. Change of continuous variables between pre-surgery and last follow-up was explored with the Wilcoxon test, due to the non-Gaussian data distribution.

Reintervention free survival time was calculated from surgery date to reintervention date, or last contact date. Reintervention-free survival was plotted according to the Kaplan–Meier method. The estimate of reintervention-free survival for any reason was calculated at six months and one year and presented as percentage and 95% confidence interval (CI).

## 3. Results

### 3.1. Selection of Study Population

A total of 37 periprosthetic femoral component fractures were identified and enrolled in the present study, following an exclusion process. The details about the flow chart of the patient selection and inclusion process are shown in Figure 1.

Demographic findings were also reported. Among the study population, two individuals had previously endured ipsilateral hip surgery. The first patient had received fixation with plates and screws due to a peritrochanteric fracture, while the second one had undergone Ganz osteotomy in the setting of hip dysplasia. Neither of these patients experienced any medical or mechanical complications following the operation, with positive indications of functional and radiographic recovery.

Demographic results are reported in Table 2. Patients’ American Society of Anesthesiology (ASA) score and Body Mass Index (BMI) are reported in Table 3. Comorbidities of the study population are reported in Table 4.

### 3.2. Characteristics of Study Population

Most of the primary THAs were uncemented (35, 95.59%) and only 2 (5.41%) employed cemented stems. All the stems employed were monoblock.

Mean LOS was 5.81 ± 1.56 days (range 4–9). The detailed analysis of the operative management is presented in Table 5.

The Kaplan–Meier analysis estimated that overall freedom from reintervention was 100% at 30 and 90 days, and would stabilize at 99% (95%CI) at 6 months. There was only one case of reintervention among the population observed, concerning a monoblock uncemented stem (Wagner Conus, Zimmer Biomet, Warsaw, Indiana, US).

Two implant-related complications were observed in the present cohort (5.41% of the population), both occurring during the first 30 days following the THA: one case of stem loosening and dislocation in a female over 65 years of age (2.70%), and one case of nonunion of the PPHF (2.70%). The first patient underwent revision arthroplasty (RA) within the first 6 months following operation, whereas the second one did not experience significant impairment in hip functionality and did not require revision surgery up to the present time.

Only one patient (2.70%) sustained medical complications within 30 days postoperatively, experiencing a sudden rise in c-reactive protein (CRP) during the 5th postoperative day, provoked by a local infection managed with antibiotic regimen administration.

One patient (2.70%) experienced an accidental fall at two months from the intervention, with no noteworthy impact on the clinical course. No patients died during the follow-up period. Patients received an average follow-up of 19.30 months (range 12–35). Table 6 lists the postoperative mechanical and medical complications.

The mean HHS and LLD were subject to significant postoperative improvement, as reported in Table 7. Radiographic examples of a successful treatment and a failure model are displayed in Figure 2 and Figure 3.

## 4. Discussion

The main finding of this retrospective study is the effectiveness of treating VB1 PPHFs that occur in primary THAs with cerclage wires with or without screws; this strategy entails a good reintervention-free survival rate at 1 year of follow up and improvement of radiographic outcomes and functional assessment.

The management of intraoperative VB1 fractures represents a significant challenge for surgeons, since their propagation in the postoperative setting could determine implant failure and reintervention. The objective of promptly treating intraoperative periprosthetic fractures is to ensure bone healing, avoid their postoperative propagation and ultimately enable full functional recovery of the hip, thereby averting further surgical interventions.

Despite their growing incidence, there is currently no established gold standard for treating VB1 IOPFFs, and surgeons often rely on their expertise in managing these cases. Previous biomechanical studies have investigated the effectiveness of treating VB1 IOPFFs with extramedullary cerclage constructs, proving their effectiveness both in axial load and torsional load testing, especially when using metallic wires with positive locking systems [19]. Additionally, the adoption of either two or three cerclage wires for the fixation of femoral periprosthetic fractures does not entail any statistically significant difference at the biomechanical level. This is because both approaches equally ensure and guarantee stability in the femur [20].

A case–control study has evidenced favorable outcomes for cerclage cabling in fractures of the calcar occurring during cementless THAs (focusing on clinical outcomes, amount of stem subsidence and implant survivorship) [21]. Furthermore, additional research has underscored the importance of precise risk assessment during preoperative planning and emphasized the significance of identifying intraoperatively the surgical steps with higher fracture risk (canal preparation and component insertion) [22].

The incidence of periprosthetic fractures is expected to increase due to the rising diffusion of THAs and an aging population. A study conducted in this field reports an incidence of femoral PPFs of 1.70% (564 events on 32.644 surgical procedures) [8].

The overall incidence of IOPFFs in the surgical unit where this study was conducted was reportedly even lower, around 1.42% (37 events on 2606 procedures, performed on 2350 patients). The main factors accounting for this result are most likely the proficiency of the surgical team, along with accurate implant selection and preoperative planning.

None of the patients included in the study population have encountered mortality within one year post surgery, aligning with the existing mortality rates observed following primary THA (0.30%) [23]. Furthermore, in our center, the awareness of possible major systemic complications after invasive procedures leads to comprehensive perioperative management, which could lead to better outcomes.

The stem mobilization rate within one year for our cohort was 2.70% (n = 1). The event was most likely due to a flawed classification of an IOPFF with stem instability as VB1, along with poor adherence of the patient to postoperative recommendations.

Certainly, the classification of IOPFFs into B1 or B2 represents a significant challenge for surgeons due to the difficulty in assessing intraoperatively the stability of the femoral component. The strategy employed by our center to evaluate stem stability in IOPFFs occurring after the definitive placement of the femoral component is to apply a vertical and rotational preloading stress. If, at this stage, the vertical and lateral offset aligns with the planned measurements, the fracture will be classified as stable.

IOPFF occurrence entailed a slight increase in the average LOS in comparison to that reported for primary THAs in the previous literature (5.81 vs. 4.00 days) [24]. Regardless, it is important to note that the calculation of LOS varies depending on the time of patients’ admission (whether it is the night before the intervention or on the same day). In the center where the present study was conducted, protocol requires admitting patients the day before the scheduled operation. Therefore, if it was calculated starting from the date of intervention, the LOS of each patient would be diminished by one day, narrowing the deviation from the values documented in the existing literature.

The incidence of IOPFFs is increasing. IOPFFs are 14 times more likely to occur within uncemented stems in females older than 65 years old, mainly during femoral broaching or stem insertion [8]. This aligns with our findings, since the majority of intraoperative Vancouver B1 fractures occurred around uncemented rather than cemented stems (95.59% vs. 5.41%). To mitigate fracture risk, cemented implants should be the primary treatment choice in elderly patients of female sex, as suggested by the Spotorno scoring system that classifies patients according to their gender, age, Singh’s Index and morpho-cortical index.

This study has several limitations. First, it is a retrospective observational study; thereby, as with any database, the quality of data and missing data may introduce errors and the sample size is limited. Additionally, as a cohort study, the absence of a control group does not allow for comparisons between cerclage cabling (with or without wires) and other therapeutic strategies. The inclusion of patients solely coming from a single center may also introduce selection bias into the analyzed data. Lastly, one limitation is represented by the reduced length of patients’ follow-up, which nevertheless was sufficient for determining the endpoints of this study since implant failure generally occurs in a brief timeframe. The long-term effects of this technique in terms of aseptic loosening remain unknown.

The main strength of this study is that, to the best of our knowledge, no gold standard exists for treating intraoperative VB1 IOPFFs in primary THAs. This is the first study focusing exclusively on the effectiveness of treating intraoperative VB1 fractures occurring during THAs with cerclage cabling. Moreover, this study relies on arthroplasties performed in a single high-volume center; thereby, the centralized patient selection enhances the homogeneity of perioperative treatment of the study population, providing a more consistent and specific analysis.

## 5. Conclusions

Cerclage wires prove to be a reliable and secure surgical method for addressing intraoperative Vancouver B1 fractures in total hip arthroplasties.The technique is associated with favorable outcomes, including optimal fracture healing, complete functional recovery and a high reintervention-free survival rate at the one year.Radiographic and clinical evaluations of patients undergoing the aforementioned surgical approach reveal a favorable postoperative clinical course, with infrequent onset complications and excellent recovery.Additional research is encouraged in order to comprehensively assess the long-term outcomes, comparative effectiveness and patient-reported experiences associated with cerclage cabling and different fixation techniques, aiming to further refine and optimize the management of intraoperative VB1 fractures.

## Figures and Tables

**Figure 1 jcm-13-00819-f001:**
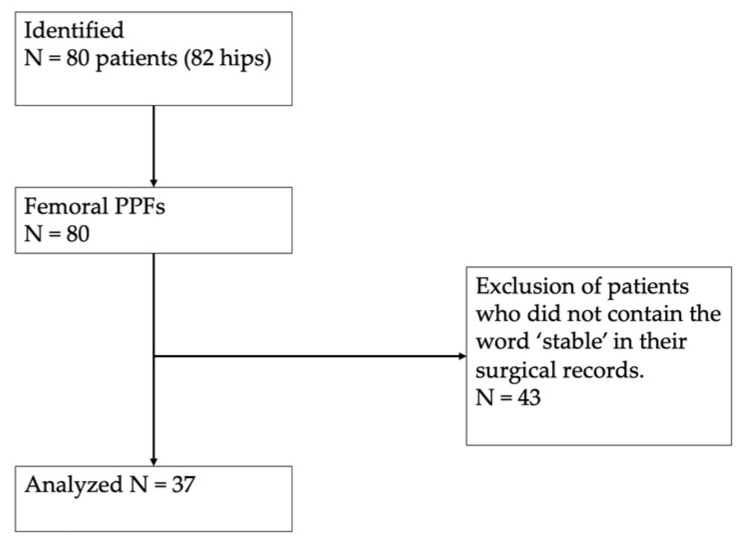
Flow chart of the patient selection and inclusion process.

**Figure 2 jcm-13-00819-f002:**
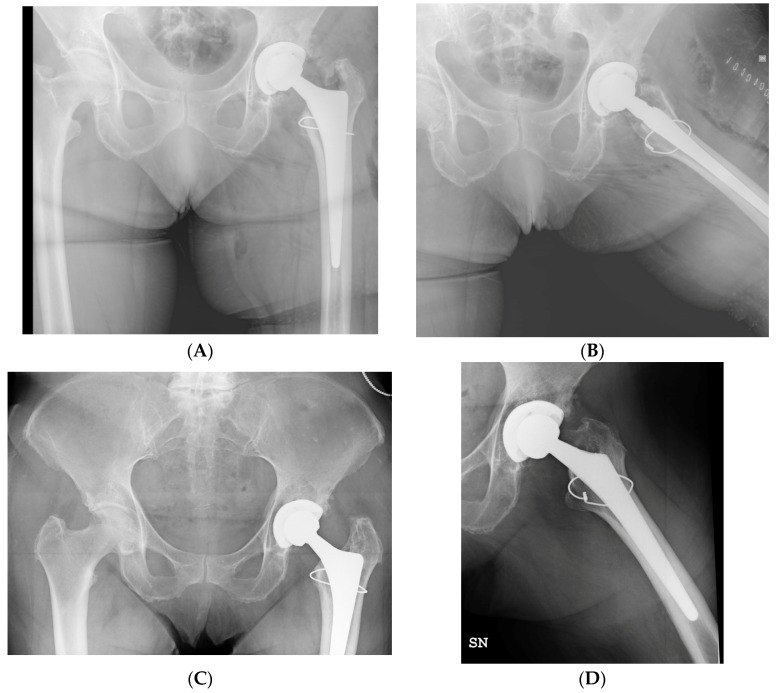
(**A**,**B**) Postoperative radiographs show the VB1 fracture of a 68-year-old female, treated with single metal cerclage cabling during primary total hip arthroplasty. (**C**,**D**) Subsequent radiographic assessments at 19 months follow-up demonstrate excellent healing of the fracture, with no indications of periosteal reaction or mobilization of the stem.

**Figure 3 jcm-13-00819-f003:**
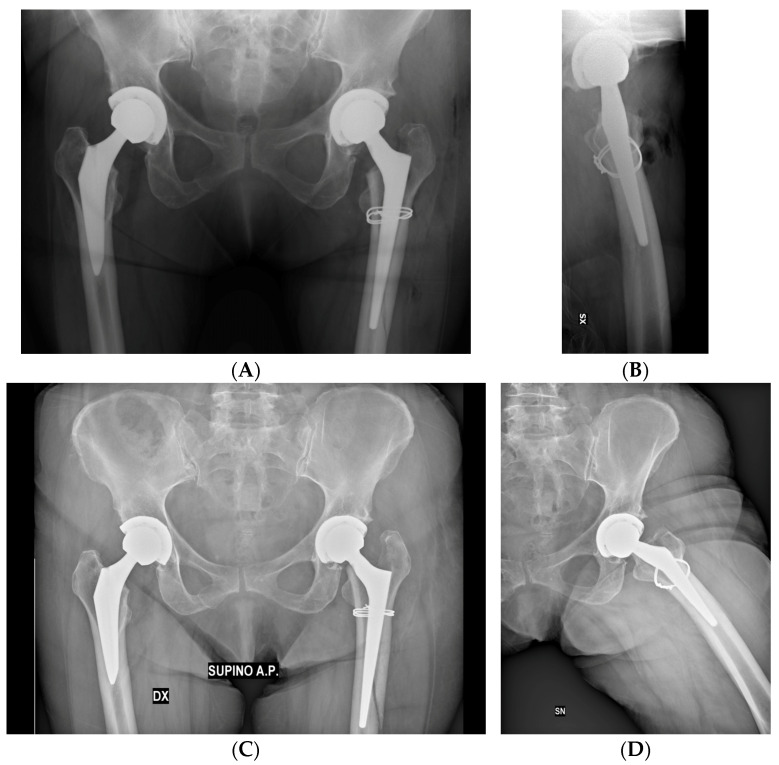
(**A**,**B**) Postoperative radiographs evidence the intraoperative VB1 fracture of a 71-year-old female, treated with double metal cabling during THA. (**C**,**D**) Follow-up radiographic evaluations of the same patient at 6 months exhibit signs of implant loosening and stem mobilization, probably caused by flawed intraoperative lesion classification and poor adherence of the patient to the recommendations given by the surgeon. Reintervention with revision arthroplasty was necessary to address this postoperative complication.

**Table 1 jcm-13-00819-t001:** The Vancouver classification system.

Classification		Fracture Location
A	AG	Greater trochanter fracture
	AL	Lesser trochanter fracture
B	B1	Fracture around the prosthesis, stem well fixed
	B2	Fracture around the prosthesis, stem loose
	B3	Fracture around the prosthesis, loose stem and poor proximal bone stock
C		Fracture distal to tip of stem

**Table 2 jcm-13-00819-t002:** Demographic results.

N	37
Age at surgery (years)	60.03 ± 15.49 (22 to 77)
Male	13 (35.14%)
Female	24 (64.86%)
Right	19 (51.35%)
Left	16 (43.24%)
Bilateral	2 (5.41%)
Previous ipsilateral hip surgery	2 (5.41%)
Active smokers	6 (16.22%)

**Table 3 jcm-13-00819-t003:** ASA score and BMI.

BMI 19–24.9 (normal weight)	17 (45.35%)
BMI 25–29.9 (overweight)	13 (35.14%)
BMI 30–34.9 (class I obesity)	7 (18.92%)
ASA score I	35 (94.59%)
ASA score II	1 (2.70%)
ASA score III	1 (2.70%)

**Table 4 jcm-13-00819-t004:** Patients’ Comorbidities.

Hypertension	13 (35.14%)
Dyslipidemia	11 (29.73%)
Aortic Aneurism	3 (8.11%)
DMT2	1 (2.70%)
Sarcoidosis	1 (2.70%)
Osteopenia/Osteoporosis	3 (2.70%)
Dysthyroidism	6 (16.22%)
Pneumological Diseases (Asthma, BPCO)	4 (10.81%)
Previous Oncological History	3 (8.11%)
Varices	1 (2.70%)

DMT2, Diabetes Mellitus Type 2.

**Table 5 jcm-13-00819-t005:** Detailed analysis of operative management.

Operating Time (Minutes)	101.68 ± 41.37 (40–237)
Patients enrolled	37 (100%)
Uncemented stems	35 (95.59%)
Zimmer GTS	12 (32.43%)
Zimmer CLS	13 (35.14%)
ATESOS pyramid	4 (10.81%)
OHST Expersus	1 (2.70%)
Zimmer Fitmore	1 (2.70%)
Wagner Conus	4 (10.81%)
Cemented stems	2 (5.41%)
MS 30	1 (2.70%)
Adler Ortho Tris	1 (2.70%)
Cerclage wires	37 (100%)
Cerclage wires with Screws	3 (8.10%)
LOS (days)	5.81 ± 1.56 (range 4–9)

**Table 6 jcm-13-00819-t006:** Mechanical and medical complications.

Mechanical Complications	
Stem loosening and mobilization	1 (2.70%)
Nonunion	1 (2.70%)
Postoperative fracture	0 (0.00%)
Nonunion	0 (0.00%)
Periprosthetic infection	0 (0.00%)
LLD	0 (0.00%)
**Medical complications**	
Superficial Wound infections	1 (2.70%)
UTI	0 (0.00%)
AF	0 (0.00%)
Pneumonia (COPD)	0 (0.00%)
Sciatic nerve palsy	0 (0.00%)
Follow-up (mean (range))	19.30 (range 12–35)

UTI, urinary tract infection; AF, atrial fibrillation; COPD, chronic obstructive pulmonary disease.

**Table 7 jcm-13-00819-t007:** Clinical and radiographic outcomes.

	Preoperative	Postoperative (at F.U.)	Delta (95% CI)	*p* Value
HHS	42.72 ± 14.37 (range 21.00–96.00)	94.40 ± 10.32 (range 56.00–100.00)	51.68 (range 4.00 to 79.00)	<0.001
LLD	−3.69 ± 6.07 (range −27.9 to 2.08) mm	0.10 ± 0.67 (range −1.07 to 1.20) mm	3.79 (range 0.21 to 26.48) mm	<0.001

## Data Availability

The data supporting reported results can be found in a repository (Zenodo).

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
