# Peer review of "Results of Utilizing Cerclage Wires in the Management of Intraoperative Vancouver B1 Fractures in Primary Total Hip Arthroplasties: A Retrospective Cohort Investigation into Clinical and Radiographic Outcomes"

_jcm, 2024, doi:10.3390/jcm13030819_

Round 1
Reviewer 1 Report
Comments and Suggestions for Authors
This manuscript investigating the effectiveness of treating VB1 PPHFs that occur in primary THAs with cerclage wires with or without screws. Due to the raising of THAs and aging of population, the incidence of periprosthetic fractures is expected to increase. This technique is linked to clinical outcomes.
Minor points:
1. The sample size is limited.
2. Figure 2 seems unnecessary.
3. Table 2 has alignment problem; It is kind of confusing.
4. Line 73: Is diaphisis a typo error of diaphysis?
Comments on the Quality of English LanguageEnglish language fine. No issues detected
Author Response
Professor Mattia Loppini
Humanitas Research Hospital, Humanitas University
Via Manzoni 56, Rozzano (Milano), Italia
mattia.loppini@hunimed.eu
Dear Reviewer 1,
I hope this letter finds you well. I am writing to express my gratitude for the effort you invested in reviewing my manuscript entitled ‘Results of utilizing cerclage wires in the management of intraoperative Vancouver B1 fractures in primary Total Hip Arthroplasties: A retrospective cohort investigation into clinical and radiographic outcomes.’, which was submitted to the Journal of Clinical Medicine with Manuscript ID jcm-2816210.
I highly appreciate your constructive feedback and your comments, which have allowed me to enhance the quality of the article, and to address potential limitations. I have carefully considered each of your recommendations and made the necessary revisions to ensure that the article meets the standards set by JCM.
Below, I provide a summary of the key revisions made in response to your feedback:
- Despite the constraints due to limited sample size, the whole analysis was performed with extreme methodological rigor. Proper statistical approaches were used to improve the robustness of the results, and extensive analyses were performed to reduce any potential biases or limits related to the sample size, guaranteeing the results' statistical validity and reproducibility. It has also been considered as a limit of the present retrospective study in the discussion section.
- Figure 2 has been removed, in accordance with the recommendations.
- The table format has been ameliorated.
- The typing error at line 82 has been corrected.
In addition to the major revisions, I have also addressed minor suggestions, such as improving the clarity of the results section, re-examining the design of the research, refining the language, and ensuring adequate formatting.
I believe the aforementioned revisions have strengthened the quality of the manuscript.
I have attached the revised paper along with a marked-up version (modifications to the manuscript are highlighted in red). I am hopeful that these changes adequately address your concerns, and that the manuscript is now suitable for publication in the Journal of Clinical Medicine.
I look forward to hearing from you soon about the status of my submission.
Thank you for your time and consideration.
Sincerely,
Sincerely,
Prof. Mattia Loppini.
Reviewer 2 Report
Comments and Suggestions for Authors
Dear Authors,
with great pleasure wI have conducted the review of this manuscript for JCM.
In brief, the introduction provides sufficient background, I consider the methods to be appropriate, but the clinical algorithm to diagnose B1# is not well described. The results are clearly presented. The discussion does not focus enough on the biomechanical concepts of stem fixation.
In my opinion, The Vancouver B1 classification is rather redundant for intraoperative fractures, as the stem per se is never firmly ingrown (= stable) immediately after implantation. In my opinion, these fractures are more likely to be B2 fractures, which in my experience would require diaphyseal fixation in at least some cases. Nevertheless, the therapeutic approach and these results are very interesting. I think it is very bold to choose a stem of the same size in this setting. It is therefore of the utmost importance for this study to clearly present the authors' decision algorithm intraoperatively – Have you classified B2# differently, have you managed B2# fractures differently?
Hence, the manuscript has some major flaws. I would reconsider the manuscript after their revisions.
In the following we address the particular points:
Abstract
Ok
Introduction
Introduction provides more than sufficient information. Objectives of this study are clearly stated.
Line 78: femoral – typo.
Ll. 74-84: This paragraph is unclear. You state that in B1# the stem is removed and cerclages are placed prior reimplantation. However, in line 82 you state that there is no gold standard. If you like to discuss different techniques at this point, it is important to be precise: When and why do you remove the stem if it is still well “fixed”? What was your algorithm? If you remove the stem, I think it is important to mention diaphyseal fixation of cementless stems to bypass the fracture zone as a therapeutic option.
Material and Methods
The methods are very precisely described.
Ll. 114-122: Remove this paragraph. It should be sufficient to cite the original publication.
Ll. 160: how many cemented and uncemented stems have been revised? Refer to the Table.
L.162: See above. Please state your algorithm: Have the cerclages been applied while the stem was in situ? When and why have you opted to implant the same stem size?
Ll.169: It is sufficient to indicate that you prescribed partial weightbearing for 3-4 weeks.
Results
The Results are clearly presented.
L.202: Table I is Vancouver classification. Please update the references to the Tables and Figures.
Discussion
The discussion is thoughtful and sound. However, it has to be modified regarding the before-mentioned critique. Furthermore, it is important to discuss the timeframe: If this procedure turns out to be entirely insufficient from a biomechanical point of view, a period of one year is certainly enough to identify implant failure. Nevertheless, it must be pointed out that the long-term effect of this technique regarding aseptic loosening remains unknown.
References
Ok
Figures and Tables
the tables must be formatted. A postoperative X-ray would certainly be interesting.
Table 1: Demographics: Can you include at least BMI, ASA, comorbidities?
Figure 1: Femural – typo.
Author Response
Professor Mattia Loppini
Humanitas Research Hospital, Humanitas University
Via Manzoni 56, Rozzano (Milano), Italia
mattia.loppini@hunimed.eu
Dear Reviewer,
I hope this letter finds you well. I am writing to express my gratitude for the effort you invested in reviewing my manuscript entitled ‘Results of utilizing cerclage wires in the management of intraoperative Vancouver B1 fractures in primary Total Hip Arthroplasties: A retrospective cohort investigation into clinical and radiographic outcomes.’, which was submitted to the Journal of Clinical Medicine with Manuscript ID jcm-2816210.
I highly appreciate your constructive feedback and your comments, which have allowed me to enhance the quality of the article, and to address potential limitations. I have carefully considered each of your recommendations and made the necessary revisions to ensure that the article meets the standards set by JCM.
Below, I provide a summary of the key revisions made in response to your feedback:
Introduction
- The typing error has been corrected according to indications.
- As per your recommendation, the algorithm employed for managing fractures has been explained with more precision under the ‘Surgical technique’ section (lines 173-182). During the procedure, the stem was removed to clear the femoral canal of eventual fracture remnants, thereby ensuring fracture reduction and avoid pressure exertion on the cortical bone. This precaution was taken to avoid bone ischemia during the application of cerclages.
Materials And Methods
- The paragraph has been removed, in accordance with the recommendations.
- Reference to Table 5 has been added as suggested (line 172).
- No, the cerclages were not applied while the stem was in place. We opted for the same stem size based on intraoperative stability of the femoral component, which was assessed manually. This decision allowed a satisfactory lateral and vertical offset, without evidence of implant subsidence or rotational instability. The decision was made with the aim of preserving the bone stock, avoiding the need for a more distal diaphyseal fixation of the stem as seen in VB2 fractures. As previously mentioned, the surgical technique paragraph has been modified in order to ensure clarity in describing the procedure.
- We chose to provide a comprehensive description of postoperative indications and rehabilitation, since they represent a pivotal factor in determining the clinical outcome investigated as well as the functional recovery of each patient.
Results
- References have been updated as recommended.
- The observation has been added to the limitations section (lines 388-389).
Figures and Tables
- In order to enhance the robustness and credibility of the study, radiographical images have been incorporated as suggested as part of the result section, including both an example of successful treatment and a failure model (Figures 2,3). Tables’ format has been ameliorated.
- Patients’ ASA and BMI and comorbidities have been added to the manuscript (Tables 3,4), in order to provide a more extensive description of the characteristics of the study population.
- The typing error in Table 1 has been addressed, according to indications.
I believe the aforementioned major revisions have strengthened the quality of the manuscript, enhancing the clarity of the whole research process. The language of the manuscript has been extensively refined and adequate formatting has been ensured.
I have attached the revised paper along with a marked-up version (modifications to the manuscript are highlighted in red). I am hopeful that these changes adequately address your concerns, and that the manuscript is now suitable for publication in the Journal of Clinical Medicine.
I look forward to hearing from you soon about the status of my submission.
Thank you for your time and consideration.
Sincerely,
Sincerely,
Prof. Mattia Loppini.
Reviewer 3 Report
Comments and Suggestions for Authors
The manuscript "Results of utilizing cerclage wires in the management of intraoperative Vancouver B1 fractures in primary Total Hip Arthroplasties: A retrospective cohort investigation into clinical and radiographic outcomes." intends to study and determine the reintervention-free survival of 84 patients affected by IOPFFs treated with cerclage wires (with or without screws) during 85 primary THAs. The study is systematic and carried in scientific methodical way.
The following comments are to be addressed.
1. The reference cited in the manuscript are old. Suggest to refer recent works in the relevant field. Suggest to cite latest reference.
2. Statistical analysis performed should be clearly explained. In case of demographic results, the patient other health conditions and complications analysis should also be recorded. This will add some important observations.
3. Suggest to include the X- rays of the patients and analyze the failure modes in a more detailed manner.
4. Strongly suggest to include the radiography images in the results and discussion section.
5. when the authors claim that "This is the first study focusing exclusively on the effectiveness of treating intraoperative VB1 fractures occurring during THAs with cerclage cabling", the authors should definitely include the radiography images which would strengthen their data and also set a benchmark standard of the future studies.
6. Please rewrite the conclusion section. list the major conclusions in bullet points.
Comments on the Quality of English Language
The general standard of English is fine. please check for some typo and also sentence construct errors.
Author Response
Professor Mattia Loppini
Humanitas Research Hospital, Humanitas University
Via Manzoni 56, Rozzano (Milano), Italia
mattia.loppini@hunimed.eu
Dear Reviewer,
I hope this letter finds you well. I am writing to express my gratitude for the effort you invested in reviewing my manuscript entitled ‘Results of utilizing cerclage wires in the management of intraoperative Vancouver B1 fractures in primary Total Hip Arthroplasties: A retrospective cohort investigation into clinical and radiographic outcomes.’, which was submitted to the Journal of Clinical Medicine with Manuscript ID jcm-2816210.
I highly appreciate your constructive feedback and your comments, which have allowed me to enhance the quality of the article, and to address potential limitations. I have carefully considered each of your recommendations and made the necessary revisions to ensure that the article meets the standards set by JCM.
Below, I provide a summary of the key revisions made in response to your feedback:
- According to the suggestion, an extensive review of existing literature has been carried out, and more recent citations have been added to the manuscript (references 12, 15, 16, 20).
- As per the recommendation, an accurate analysis of the demographic data relative to the study population has been performed, leading to the inclusion of prior ipsilateral hip surgeries into the demographic results (lines 216-221).
3, 4, 5. In order to enhance the robustness and credibility of the study, radiographical images have been incorporated as suggested as part of the result section, including both an example of successful treatment and a failure model (figures 2,3).
- The conclusion section has been accurately revised and rewritten (lines 399-410).
In addition to the major revisions, I have also addressed minor suggestions, such as improving the clarity of the methods section, re-examining the design of the research, refining the language, and ensuring adequate formatting.
I believe the aforementioned revisions have strengthened the quality of the manuscript.
I have attached the revised paper along with a marked-up version (modifications to the manuscript are highlighted in red). I am hopeful that these changes adequately address your concerns, and that the manuscript is now suitable for publication in the Journal of Clinical Medicine.
I look forward to hearing from you soon about the status of my submission.
Thank you for your time and consideration.
Sincerely,
Prof. Mattia Loppini
Round 2
Reviewer 2 Report
Comments and Suggestions for Authors
Dear Prof. Loppini,
with great pleasure I have read your response! You have sufficiently addressed all of my concerns. However, I am still skeptical about the distinction between Vancouver B1 and B2 fractures intraoperatively. This is surely explained by the fact that we have witnessed numerous subsidences in supposed B1 fractures at our center. The more impressive are therefore your results.
You address this issue as an example in Fig. 3. I would appreciate it if you could add a sentence on the difficult differentiation between Vancoucer B1 and B2 fractures in the discussion. As a reader, I would be particularly interested to know: How firmly do you investigate vertical and rotational stability? Do you have the same approach in intraoperative, reducable B2 fractures? When do you consider a diaphyseal fixation?
Author Response
Professor Mattia Loppini
Humanitas Research Hospital, Humanitas University
Via Manzoni 56, Rozzano (Milano), Italia
mattia.loppini@hunimed.eu
Dear Reviewer,
I hope this letter finds you well. I am writing to express my gratitude for the time you invested in reviewing my manuscript entitled ‘Results of utilizing cerclage wires in the management of intraoperative Vancouver B1 fractures in primary Total Hip Arthroplasties: A retrospective cohort investigation into clinical and radiographic outcomes.’, which was submitted to the Journal of Clinical Medicine with Manuscript ID jcm-2816210.
I have carefully considered each of your recommendations and made the necessary revisions to ensure that the article meets the standards set by JCM.
Below, I provide a summary of the key revisions made in response to your feedback:
- A paragraph stating the difficulty in classifying intraoperative periprosthetic fractures into Vancouver B1 and Vancouver B2 has been added to the manuscript in the discussion section (lines 367-372), along with a brief explanation of their intraoperative management in the surgical technique (177-178).
- The strategy employed by out center to evaluate the stability of IOPFFs occurring after the definitive placement of the femoral component is applying a vertical and rotational preloading stress. If, at this stage, the vertical (distance between lesser trochanter and center of rotation) and lateral offset (distance between greater trochanter and center of rotation) aligns with the planned measurements, the fracture will be classified as stable.
3,4. Concerning Vancouver B2 unstable fractures, also in the case of reducible ones, a revision stem is employed in order to achieve diaphyseal fixation and provide stability to the implant.
I have attached the revised paper along with a marked-up version (the additional paragraph is highlighted in red). I am hopeful that these changes adequately address your concerns, and that the manuscript is now suitable for publication in the Journal of Clinical Medicine.
I look forward to hearing from you soon about the status of my submission.
Thank you for your time and consideration.
Sincerely,
Prof. Mattia Loppini
